# An Observational Study to Assess the Effectiveness of 4CMenB against Meningococcal Disease and Carriage and Gonorrhea in Adolescents in the Northern Territory, Australia—Study Protocol

**DOI:** 10.3390/vaccines10020309

**Published:** 2022-02-16

**Authors:** Helen S. Marshall, Prabha H. Andraweera, James Ward, John Kaldor, Ross Andrews, Kristine Macartney, Peter Richmond, Vicki Krause, Ann Koehler, David Whiley, Lynne Giles, Rosalind Webby, Heather D’Antoine, Jonathan Karnon, Rob Baird, Andrew Lawrence, Helen Petousis-Harris, Philippe De Wals, Belinda Greenwood-Smith, Michael Binks, Lisa Whop

**Affiliations:** 1Vaccinology and Immunology Research Trials Unit, Women’s and Children’s Health Network, Adelaide, SA 5006, Australia; prabha.andraweera@adelaide.edu.au; 2Adelaide Medical School and Robinson Research Institute, The University of Adelaide, Adelaide, SA 5005, Australia; 3UQ Poche Centre, The University of Queensland, Saint Lucia, QLD 4072, Australia; james.ward@uq.edu.au; 4The Kirby Institute, The University of New South Wales, Sydney, NSW 2052, Australia; jkaldor@kirby.unsw.edu.au; 5Menzies School of Health Research, Casuarina, NT 0811, Australia; ross.andrews@menzies.edu.au (R.A.); heather.dantoine@menzies.edu.au (H.D.); michael.binks@menzies.edu.au (M.B.); lisa.whop@anu.edu.au (L.W.); 6Discipline of Paediatrics and Child Health, The University of Sydney, Sydney, NSW 2006, Australia; kristine.macartney@health.nsw.gov.au; 7Telethon Kids Institute and The University of Western Australia, Crawley, WA 6009, Australia; peter.richmond@uwa.edu.au; 8Centre for Disease Control, Department of Health, Northern Territory Government, Darwin, NT 0810, Australia; vicki.krause@nt.gov.au (V.K.); rosalind.webby@nt.gov.au (R.W.); belinda.greenwoodsmith@nt.gov.au (B.G.-S.); 9Communicable Disease Control Branch, SA Health, South Australian Government, Adelaide, SA 5000, Australia; ann.koehler@sa.gov.au; 10UQ Centre for Clinical Research, The University of Queensland and Pathology Queensland, Saint Lucia, QLD 4072, Australia; d.whiley@uq.edu.au; 11School of Public Health, The University of Adelaide, Adelaide, SA 5005, Australia; lynne.giles@adelaide.edu.au; 12College of Medicine and Public Health, Flinders University, Bedford Park, SA 5001, Australia; jon.karnon@flinders.edu.au; 13Northern Territory Pathology, Department of Health, Northern Territory Government, Casuarina, NT 0811, Australia; rob.baird@nt.gov.au; 14SA Pathology, Department of Health, South Australian Government, Adelaide, SA 5000, Australia; andrew.lawrence@sa.gov.au; 15Department of General Practice and Primary Health Care, The University of Auckland, Auckland 1023, New Zealand; h.petousis-harris@auckland.ac.nz; 16Faculty of Medicine, The Laval University, Quebec, QC G1V 0A6, Canada; philippe.dewals@criucpq.ulaval.ca

**Keywords:** vaccination, 4CMenB, vaccine, meningitis, meningococcal disease, gonorrhoea

## Abstract

Invasive meningococcal disease (IMD) causes significant morbidity and mortality worldwide with serogroup B being the predominant serogroup in Australia and other countries for the past few decades. The licensed 4CMenB vaccine is effective in preventing meningococcal B disease. Emerging evidence suggests that although 4CMenB impact on carriage is limited, it may be effective against gonorrhoea due to genetic similarities between *Neisseria meningitidis and Neisseria gonorrhoeae*. This study protocol describes an observational study that will assess the effect of the 4CMenB vaccine against meningococcal carriage, IMD and gonorrhoea among adolescents in the Northern Territory (NT). All 14–19-year-olds residing in the NT with no contraindication for 4CMenB vaccine will be eligible to participate in this cohort study. Following consent, two doses of 4CMenB vaccine will be administered two months apart. An oropharyngeal swab will be collected at baseline and 12 months to detect pharyngeal carriage of *Neisseria meningitidis* by PCR. The main methodological approaches to assess the effect of 4CMenB involve a nested case control analysis and screening method to assess vaccine effectiveness and an Interrupted Time Series regression analysis to assess vaccine impact. Research ethics approvals have been obtained from Menzies and Central Australian Human Research Ethics Committees and the Western Australian Aboriginal Health Ethics Committee. Results will be provided in culturally appropriate formats for NT remote and regional communities and published in international peer reviewed journals. ClinicalTrials.gov Identifier: NCT04398849.

## 1. Introduction

Invasive meningococcal disease (IMD) caused by *Neisseria meningitidis* is a rare but serious infection and an important cause of mortality worldwide [1,2]. Clinically it presents as meningitis and or septicaemia, with a case fatality rate between 4.1–20.0% [3]. *N. meningitidis* is classified into serogroups based on the biochemical composition of the capsular polysaccharide, with serogroup B causing the majority of IMD in Europe, Australia, New Zealand and much of America [4].

Exposure to *N. meningitidis* is common in the general population, resulting in asymptomatic oropharyngeal carriage that may be transient, or long-term [5]. Age influences carriage, with an increase from 15 years to a peak at around 19 years [6]. The United Kingdom introduced the meningococcal B vaccine (4CMenB^®^, Bexsero^®^) into its national immunization program for infants and toddlers in 2015 and the vaccine was shown to protect children under the age of 2 years from IMD [7]. However, 4CMenB does not have significant effects on carriage of disease-causing meningococci, including group B [8].

Emerging evidence suggests that 4CMenB may protect against gonorrhoea, a sexually transmissible infection (STI) which is identified by the World health Organisation as a public health priority. An estimated 106 million cases of gonorrhoea are diagnosed worldwide each year with the number increasing in recent years [9]. Gonococcal infections commonly present as urethritis in males and cervicitis in females, but mucosal infections of rectum, pharynx and eye frequently occur [10,11,12]. In addition, gonococcal infections can lead to pelvic inflammatory disease, adverse pregnancy outcomes, infertility and neonatal complications. In Australia, gonorrhoea notifications increased by 80% over a 5-year period from 2013 to 2017 (from 65.5 to 118 per 100,000) [13]. The Northern Territory (NT) has the highest rate of gonorrhoea notifications in Australia with 686.6 per 100,000 reported in 2017 [14]. The gonorrhoea notification rate in the Aboriginal and Torres Strait Islander population was 6.6 times higher than the non-Indigenous population in 2017 (627.5 per 100,000 versus 95.6 per 100,000) [13]. A recent study on potential disseminated gonococcal infection between January 2010 and September 2018 reported 106 cases in the Top End of the NT. Of these, 94 (88.7%) were among Indigenous Australians and the highest incidence was in the 15–19 year old age group [15]. Variations exist across regions within the NT, which is divided into two major areas; the Top End which covers 516,945 square kilometres and a population of approximately 250,000 and Central Australia, which covers 830,220 square kilometres and a population of approximately 41,000. The Central Australian region communities have much higher notification rates than the Top End communities.

The effectiveness of antibiotics for gonorrhoea has been compromised due to emerging resistance to broad-spectrum Cephalosporins, and development of an effective gonorrhoea vaccine has been challenging. However, surveillance reports from Cuba, New Zealand and Canada showed a decline in gonorrhoea incidence following vaccination against *N. meningitidis* [10,11,12,16]. A recent transmission-dynamic model showed that a vaccine conferring 31% protection (as estimated for the MeNZB) for 2–4 years could reduce the incidence of gonorrhoea by 45% by 2030 [17].

In addition to having the highest rates of gonorrhoea, the NT also has the highest rate of IMD in Australia with 2.0 per 100,000 reported in the first 6 months of 2019 [18]. Carriage prevalence of *N. meningitidis* is higher in Aboriginal people than in non-Aboriginal people [8]. As the NT has the highest burden of both gonorrhoea and meningococcal disease in Australia, this is an important setting to conduct the proposed study.

The aim of this study is to implement a 4CMenB vaccination program for young people aged 14–19 years of age, residing in the NT and to evaluate the effectiveness of the vaccine on gonorrhoea, meningococcal carriage and IMD.

## 2. Materials and Methods

Exclusion criteria include anaphylaxis following any component of the Bexsero^®^ vaccine, previous receipt of the MenB vaccine, Bexsero^®^ (previous receipt of MenNZB^TM^ is allowed), known pregnancy, clinical conditions representing a contraindication to intramuscular vaccination and venepuncture. A temporary exclusion of a 14-day interval between receipt of study vaccine and COVID-19 vaccines applies. Eligible 14–19-year-olds can participate in the study via schools or via government and non-government immunisation clinics including Aboriginal Community Controlled Health Organisations (ACCHO). The 14–19-year-old age group is selected to enable vaccination prior to rapid carriage and STI acquisition. Conducting the study in schools provides an opportunity to use the existing school immunization program and infrastructure in a closed accessible environment with potential for the highest rates of participation. The objectives and outcome measures are detailed in Table 1.

### 2.1. Study Population

All 14–19-year-olds residing in the NT are eligible to participate. It is expected that a large number of the participants will be Aboriginal and Torres Strait Islander young people comprising 40% of the population of the NT, the highest proportion of Indigenous people in any Australian state or territory Australian Bureau of Statistics. Estimates of Aboriginal and Torres Strait Islander Australians [19]. Of the 14–19-year-old Aboriginal and Torres Strait Islander Australians residing in the NT, approximately, 52% live in outer regional areas, 19% live in remote and 29% live in very remote regions Australian Bureau of Statistics. Estimates of Aboriginal and Torres Strait Islander Australians [19].

### 2.2. Study Processes

Immunisation providers will receive training in all study procedures, including informed consent, collection of posterior oropharyngeal swabs and 4CMenB vaccination.

Those participating via the school program will require written/electronic informed consent from the parent/guardian. Those participating via health clinics will be able to provide written consent for themselves if they are aged 16 years and above and parental/guardian consent will be required for those under 16 years. Participants and parents are provided with the contact details of the study coordinators to obtain any additional information that they may require prior to or after receiving the vaccination.

Participation includes three visits (Figure 1). During visit one, the participants will be asked to complete a one-page de-identified questionnaire to obtain information on characteristics that are known to be relevant to carriage of *N. meningitidis* including smoking/tobacco history, household size, recent respiratory tract infections and recent use of antibiotics. The immunisation providers will then obtain an oropharyngeal swab which will be placed immediately in STGG transport medium (skim milk, tryptone, glucose and glycerine; Thermo-Fisher Scientific Australia) [20] and transported to NT Pathology [21]. The immunisation provider will then administer the first dose of the licensed 4CMenB vaccine. The second visit will be scheduled a minimum of 2 months after the first visit and an SMS reminder will be sent 2 days prior to the scheduled date to notify parents/guardians/participants of the follow up visit. During the second visit, the participants will receive the second dose of the 4CMenB vaccine. The required minimal time interval between the 4CMenB vaccine and other vaccines including the COVID-19 vaccine will be maintained based on prevailing guidelines. The third visit will be scheduled 12 months after the first visit and a similar SMS reminder will be sent. During the third visit, the participants will be asked to complete the same study questionnaire that was completed at visit one and the second posterior oropharyngeal swab will be obtained. All participants will receive a $20 value voucher at visits 1 and 3 to reimburse them for their time.

All collected data including consent forms, questionnaires and swab analysis results will be securely stored on a password-protected database held by the Adelaide Health Technology Assessment, The University of Adelaide. Range and logic checks will be performed on all collected data.

### 2.3. Patient and Public Involvement

The study materials will be reviewed by a Youth Advisory Group and an Aboriginal Advisory Group at several stages. Public, Independent, Catholic, Christian and Lutheran schools in the NT will provide information to parents/guardians and students and support the roll out of the study in schools. The ACCHOs will provide information about the study to their communities and support the roll out in clinics. A multimedia communication strategy has been developed by a public relations/communications company in collaboration with the media team of the University of Adelaide. Key strategies include website development (B Part of it NT. Available online: bpartofitnt.com.au (accessed on 17 January 2022)), brand identity ‘B Part of it NT’, a Facebook page, advertising and creating supporting material in English and up to 10 Aboriginal languages to be used in remote settings, a flip chart explaining the study in a simplified version to be used alongside the participant information sheets, ambassador engagement, public relations and management and media training. Participants and parents are encouraged to provide feedback on the study using the B Part of it NT Facebook page.

### 2.4. Ethics and Dissemination

The study has been approved by the Human Research Ethics Committee (HREC) of the Northern Territory Department of Health and Menzies School of Health Research (Reference Number: 2019-3507), the Central Australian HREC (Reference Number: 2019-3524) and the Western Australian Aboriginal Health Ethics Committee (Reference Number: 996). The progress of the study will be communicated to all stakeholders via monthly newsletters. The final study results will be provided to participants, parents/guardians, stakeholders and the general public via media including television, radio and print media. Results will be provided in culturally appropriate ways as advised by the Aboriginal advisory Group and Youth advisory group including use of media. The results will be presented at local, national and international scientific meetings and published in peer reviewed journals.

### 2.5. Study Safety Monitoring and Surveillance

Vaccine safety and adverse events will be routinely monitored through an enhanced passive surveillance system used for timely detection of adverse events following immunisation through the NT Department of Health, Centre for Disease Control (CDC). Participants/parents are given the contact details of the Immunisation Unit of the NT Centre for Disease Control and advised to report any AEFI to the Immunisation Unit. Serious adverse events (SAEs) which are considered to be related to administration of 4CMenB will be reported to the HRECs, the study sponsor, the Therapeutic Goods Administration (Australian Government) and the vaccine manufacturer within 72 h of the site becoming aware of the SAE. A study vaccine safety committee comprising independent vaccine safety experts has been established and will review all study-related vaccine safety data in accordance with a vaccine safety surveillance protocol. Monthly summaries of all reported adverse events will be provided to the vaccine safety committee and the vaccine manufacturer.

### 2.6. Training of Immunisation Providers

Immunisation nurses will be trained in obtaining posterior oropharyngeal swabs to ensure a standardised collection technique. Training manuals and a training video including standard swab collection will be available at all sites for continuous training. Schools and clinics will be randomly selected for monitoring of protocol-related study processes including consent and technique of obtaining the throat swab.

### 2.7. Laboratory Processes

On receipt of samples at NT Pathology, DNA will be extracted using automated extraction on the Roche MagnaPure system and subjected to PCR screening for the presence of specific meningococcal DNA (using PorA gene detection). Any sample yielding a positive PCR will be identified and cultured for *Neisseria* species on selective and non-selective agar and incubated overnight in CO_2_ at 35 °C. Plates will be examined daily for isolates up to 72 h. *N. meningitides* will be identified by standard diagnostic laboratory bacteriological methods using oxidase reaction and MALDI with further PCR testing to determine the capsular group (A, B, C, W, X, Y). Genomic DNA will be extracted and sequenced via the Illumina platform. Raw reads will undergo quality control and trimming to ensure only high-quality reads are analysed. Relevant typing data, resistance markers, and phylogenetic analyses will be carried out using several software tools [22].

### 2.8. Data and Statistical Analysis Plan

Basic demographic data and risk factors for meningococcal disease (from the questionnaire), laboratory results and vaccination data will be entered into the study database. IMD and gonorrhoea rates will be obtained from routine surveillance data from the CDC, NT. For the case–control and screening method analyses, data custodians from AIR and CDC NT will link participant records and assign unique study IDs prior to sending data to the study team. No identifying information will be stored in the main study database. Infection rates for IMD and gonorrhoea for various demographic subgroups will be estimated from CDC NT surveillance data, and a many-to-one merge will be used to link background rates to individual records, so as to adjust for these rates in the subsequent analyses.

There are some limitations for observational studies with potential for bias and confounding. Some of the potential sources of bias identified include, any changes to the current practice of notifications over the study period, differences in case ascertainment due to variation in extent of screening and contact tracing, differences in healthcare seeking behaviour for STI treatment and vaccination, extent of other STI prevention/health promotion programs offered in the region, missing data, misclassification of the outcome, differences in lab diagnostics and those lost to follow up during the study period.

To partly mitigate the effects of some of the potential biases, different study designs will be utilized in the investigation of the research hypotheses. This approach will aid in triangulation of findings and strengthen the conclusions that will be drawn from the study [23]. Since there is considerable debate about the circumstances under which causality can be tested or assumed, we will use two different approaches with differing and unrelated key sources of potential biases in assessing vaccine effectiveness. The two are the screening approach and the case–control approach. In the screening approach, results are compared between different populations in different contexts (cross-context comparisons). In the case–control approach, two different control groups will be used, namely controls from the STI notification (Chlamydia) database and the Australian Immunisation register (AIR). These approaches will not give the same estimates of the causal effect unless all analyses are unbiased.

### 2.9. Sample Size

For ethical and equity reasons, all 14–19-year-olds in the NT in 2021–2022 will be eligible to be offered 4CMenB. The estimated eligible population is 14,077, of whom 5621 or 40% are Aboriginal or Torres Strait Islander (June 2016 estimates). We conservatively anticipate 40–50% participation of this population group and up to 30% non-return at 12 months due to factors such as interstate and intra-territory travel.

#### 2.9.1. Gonorrhoea Infection Power Calculations

Case–control study: To assess the vaccine effectiveness of 4CMenB on gonorrhoea, assuming 1350 cases of gonorrhoea among 15–19-year-olds in the NT (Australian Bureau of Statistics) over a three year period (i.e., 450 per annum), a ratio of 3 matched controls to 1 case and assuming 50% of the eligible control population vaccinated with a correlation of 0.1, the study has at least 90% power at alpha = 0.05 to detect vaccine effectiveness of at least 20% (i.e., OR = 0.8 or less) and allowing for correlation between cases and controls on basis of 0.2. With this estimated number of cases and controls, the study is adequately powered even if 40% of the eligible population is vaccinated.

Interrupted time series study: A power simulation study that assumed three years of monthly observations post-implementation in each of the vaccinated and unvaccinated groups, constant gonorrhoea counts over time in the unvaccinated group (i.e., ~450/12 = 37.5 cases per month, so µ = 3.62), and a Poisson distribution of counts was conducted. Using the method of Feiveson with 1000 runs, the power to detect a coefficient of −0.008 in the vaccinated group as significant is 0.84 (alpha = 0.05, two-tailed test). This equates to a 9.6% reduction in cases in the vaccinated group relative to unvaccinated group (from 450 to 407 cases p.a.), assuming the two groups are of equal size.

#### 2.9.2. Meningococcal Carriage Power Calculations and Impact on Disease

Assuming an overall carriage prevalence at baseline of 6.8% (B Part of it study, unpublished data), a sample size of 5239 is required in order to have 80% power to detect a 20% reduction at 12 months (alpha = 0.05, two-tailed test) or 7011 participants to have 90% power. We will recruit at least 5239 and up to 7011 participants during the study. This calculation incorporates a correlation between the baseline and 12-month measures of 0.25 (‘B Part of It’ study) and 30% loss to follow-up.

### 2.10. Statistical Analyses

All analyses will be performed according to a pre-specified statistical analysis plan using Stata version 15.

## 3. Primary Objectives

### 3.1. Evaluation of the Effectiveness of 4CMenB Vaccine against Gonorrhoea in 15 to 19 Year Olds

The primary analysis on vaccine effectiveness (VE) will be performed using a case–control method using controls from the sexually transmitted infections (STI) notification database for chlamydia. A case will be defined as any laboratory confirmed case of gonorrhoea among 15–19-year-olds notified to the CDC. For each case, up to four controls will be randomly sampled from a de-identified dataset of individual records extracted from the Chlamydia notification database of the CDC. Although increasing the number of controls per case increases statistical power, beyond a ratio of four controls per case only small improvements in power are expected [24,25]. A secondary case–control analysis will be performed using controls from the Australian Immunisation register (AIR). For each case, up to 20 controls will be randomly sampled from a de-identified dataset of individual records extracted from the AIR, following restriction of the database to the NT to ensure sampling from the same population. A previous similar study on assessing the effectiveness of the varicella vaccine in Australia has shown maximum precision using 20 controls per case [26]. Controls from both AIR and STI databases will be matched to cases by date of birth ± 4 weeks. Vaccination status of controls will be ascertained from the AIR after selection and vaccination considered valid if received ≥3 months before the date of laboratory confirmation of gonorrhoea in the matched case. This method of control ascertainment has been used previously to assess VE of *Haemophilus influenzae* serotype B, pertussis, varicella and measles vaccines in Australia [26]. Conditional logistic regression models will be generated in Stata to estimate VE.

VE will also be estimated using the screening method for 15–19-year-olds with laboratory-confirmed gonorrhoea within 3 months of receiving two doses of the 4CMenB vaccine. The comparator group will include all 15–19-year-olds in the NT. Using the screening method, VE will be estimated as $1-\frac{PCV(1-PCV)}{PPV(1-PPV)}$ where *PCV* is the proportion of vaccinated MenB cases and *PPV* is the vaccine coverage in the age group being analysed [27]. Vaccine coverage data for each age group will be obtained from the AIR.

### 3.2. Estimation of the Effect of 4CMenB on Carriage of all Neisseria Meningitidis in 14 to 19 Year Olds (12 Months Compared to Baseline)

Carriage prevalence of all *N. meningitidePPs* genogroups detected by PCR at 12 months will be compared with the results obtained at baseline using logistic regression. The difference in carriage at 12 months compared to baseline will be expressed as an odds ratio (OR) with 95% CI. Adjustments will be made for gender, age, smoking, ethnicity, remoteness area, household size (number of persons/room) as available and appropriate.

## 4. Secondary Objectives

### 4.1. Gonorrhoea

Vaccine impact on gonorrhoea will be estimated using surveillance data from the CDC for laboratory-confirmed gonorrhoea cases, comparing six pre-vaccine surveillance years to three post-vaccination years. Interrupted time series analysis will be used to evaluate the impact through regression modelling. To estimate impact in each age group (15, 16, 17, 18, 19 years), incidence rate ratios will be estimated by comparing case numbers in the post-vaccination period with cases in the equivalent age cohort during the six pre-vaccination years. To take into account any changes over time unrelated to 4CMenB vaccination, the IRRs will be adjusted using the relevant annual gonorrhoea incidence in all adults who were not in the vaccine-eligible age cohorts.

### 4.2. N. meningitidis Carriage

Carriage prevalence of all and each disease-causing genogroups of *N. meningitidis* (A, B, C, D, E, X, W, Y) and non-groupable *N. meningitidis* among 14–19-year-olds detected by PCR at 12 months will be compared with the results obtained at baseline using logistic regression. The difference in carriage at 12 months compared to baseline will be expressed as an odds ratio (OR) with 95% CI. Adjustments will be made for gender, age, smoking, ethnicity, remoteness area, household size (number of persons/room) as available and appropriate.

### 4.3. Meningococcal Disease

Evaluation of the vaccine impact and effectiveness of 4CMenB administered to young adults on group B IMD in 14- to 19-year-olds.

Vaccine impact on IMD will be estimated using surveillance data from CDC for laboratory-confirmed IMD cases, comparing six pre-vaccine surveillance years to three post-vaccination years as described above for gonorrhoea. Vaccine effectiveness (VE) will be evaluated using screening and case–control methods as described for gonorrhoea. For the case–control analysis, controls will be selected from AIR (20 controls per case). Controls will be matched to cases by date of birth±4 weeks. Vaccination status of controls will be ascertained from the AIR after selection and vaccination considered valid if received ≥14 days before the date of laboratory confirmation of IMD in the matched case.

### 4.4. Risk Factors

Associations between risk factors and gonorrhoea or carriage of *N. meningitidis* will be reported using descriptive statistics.

### 4.5. Analyses of Data Combined South Australian and NT Cohorts

Although not available via the National Immunisation Programme, in 2018, the South Australian State Government introduced the 4CMenB vaccine to the State Immunisation schedule for children and young people in South Australia. The adolescent program commenced February 2019 and the effectiveness of the program is currently being evaluated [28]. We will combine data from the NT study with data from SA 4CMenB adolescent program and B part of it Study conducted in SA for meningococcal disease and gonorrhoea outcomes.

### 4.6. Cost Effectiveness

A decision analytic model will be used to assess the cost-effectiveness of the 4CMenB vaccine against IMD and gonorrhoea. The model will represent the costs of delivering a vaccination program, the expected incidence of IMD and gonorrhoea with and without a vaccination program and the cost, quality of life and mortality effects of incident IMD and gonorrhoea.

A systematic review has already been conducted that informed the development of a model to evaluate the clinical and financial burden of IMD [3,29]. Costs associated with the management of these infections will be estimated using Australian hospital costing data. These will be combined with literature-based estimates of the broader costs associated with these infections, as well as quality of life and mortality effects to estimate Quality Adjusted Life Year (QALY) gains [30].

In the base case, future costs will be discounted to their present value at 5% annually and the healthcare system perspective will be employed as recommended by Australian guidelines [31]. Discount rates of 0% and 3.5% will be considered in the sensitivity analyses. The healthcare system perspective captures direct medical costs associated with the provision of a vaccination program, as well as with the treatment of IMD and gonorrhoea. To estimate indirect costs, two approaches will be used: human capital (HC) [32] and friction cost (FC) methods [33]. The HC method estimates the reduction in gross earnings due to morbidity and/or premature mortality [30]. The FC method only considers the time span employers need to restore the initial production level [34].

Incremental cost-effectiveness ratios (ICERs) will be estimated using the alternative VE analysis methods and sensitivity analyses will be undertaken to describe the effects of uncertainty around baseline incidence of IMD and gonorrhoea, VE, and the costs and consequences of IMD and gonorrhoea. Threshold analyses on the vaccine purchase price required to achieve alternative incremental costs per QALY gained of a routine immunisation program compared to no program will inform the real-world value of the immunisation program.

## 5. Discussion

Evidence for a protective effect of a meningococcal vaccine against gonorrhoea was first reported in Cuba, with a rapid decline in the incidence of gonorrhoea notifications following a vaccination campaign with VA-MENGOC-BC from 1988–1990 [10,11]. Subsequently, it was shown that the MeNZB OMV vaccine introduced in New Zealand in 2004 was also associated with reduced risk of gonorrhoea in those aged 15–30 years immunised in 2004–06 and followed to 2014 [12]. Further evidence comes from a study conducted in Quebec, Canada that analysed the cases of gonorrhoea notifications during pre- and post-vaccination periods of a targeted 4CMenB immunisation [16]. Gonorrhoea notifications among individuals aged 14–20 years were shown to decrease by 59% during the post-vaccination period whereas the notifications in those 21 years and older were shown to increase, but confidence intervals were wide (95% CI: −22% to 84%; *p* = 0.1) due to the small case numbers (average 22 per year) [16]. The authors hypothesised that cross-protection occurred despite the difference in disease manifestation, because *N. meningitidis* and *N. gonorrhoeae* share 80–90% genetic homology in primary sequences [35]. Also, recently presented data show high levels of anti-gonococcal antibodies generated in adults vaccinated with 4CMenB, which may explain cross-protection against gonorrhoea [36]. A recent review by Garcia et al. provides a comprehensive summary of the potential effect of the 4CMenB vaccine against gonorrhoea [37].

At present, there are no data from randomized controlled trials on the effectiveness of meningococcal vaccines against gonorrhoea. However, data will be available in the near future from a few ongoing clinical studies from several countries (ClinicalTrials.gov:NCT04415424, NCT04350138, NCT04398849 and Marshall et al. [38]). The proposed evaluation will provide a unique opportunity to confirm and further refine the findings from studies conducted in New Zealand and Canada, in an Australian population. Overall, the results will be of international significance to other countries that are considering implementation of a MenB vaccine program in high-risk groups.

## Figures and Tables

**Figure 1 vaccines-10-00309-f001:**
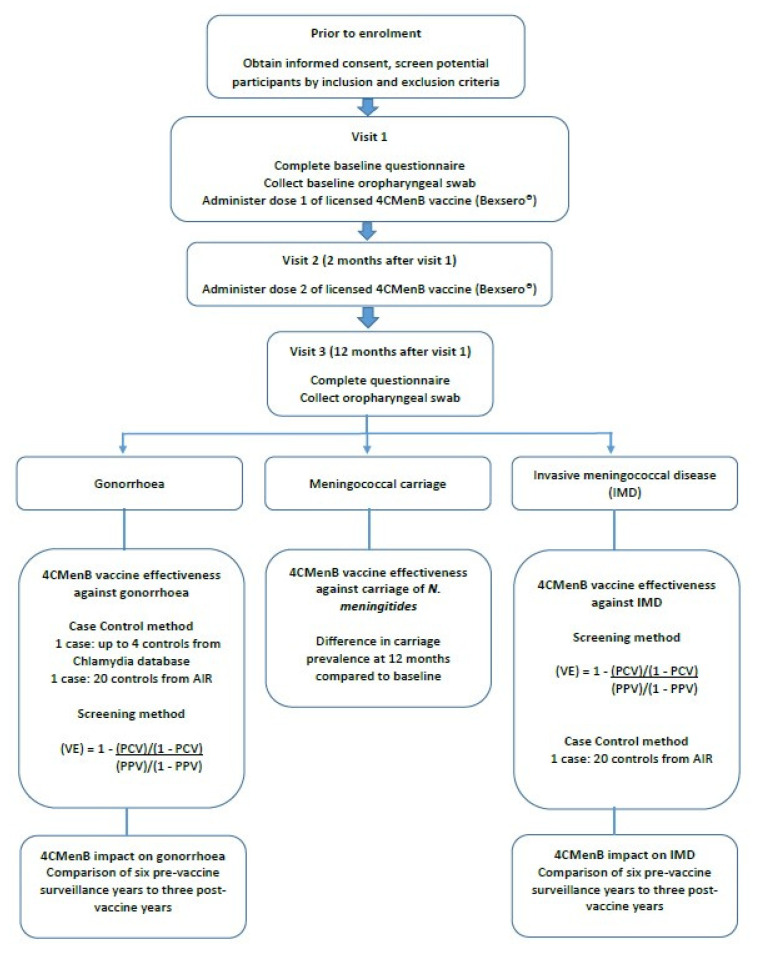
Study design.

**Table 1 vaccines-10-00309-t001:** Objectives and outcome measures.

Objective	Outcome and Outcome Measure
Primary	
Estimate the effectiveness of 4CMenB against gonorrhoea in 15- to 19-year-olds.	Primary analysis
	4CMenB vaccination status in the population with gonorrhoea, compared to randomly selected Chlamydia controls (Case–control method).
	Other analyses
	4CMenB vaccination status in the population with gonorrhoea, compared to randomly selected controls from the Australian Immunisation Register (Case–control method).
	4CMenB vaccination status in the population with gonorrhoea, compared to the general population (Screening method).
2. Estimate the effect of 4CMenB on carriage of all *Neisseria meningitidis* in 14- to 19-year-olds (12 months compared to baseline).	*N.* meningitidis carriage as measured by PCR at baseline compared to 12 months.
Secondary	
Gonorrhoea	
3. Estimate the vaccine impact of 4CMenB on gonorrhoea notification rates in 15- to 19-year-olds (difference in gonorrhoea notification rates before and after the introduction of the 4CmenB vaccine).	All laboratory confirmed notifications of gonorrhoea in the six years preceding 4CMenB vaccination compared to three years post vaccination (Interrupted time series analysis).
	All laboratory confirmed notifications of gonorrhoea in 15–19-year-olds in the vaccinated population compared to the unvaccinated population.
4. Estimate 4CMenB vaccine impact and effectiveness against gonorrhoea in males and females.	All laboratory confirmed notifications of gonorrhoea in the vaccinated population compared to the unvaccinated population stratified by gender.
5. Estimate 4CMenB vaccine impact and effectiveness against gonorrhoea by location (regional, remote, very remote).	All laboratory confirmed notifications of gonorrhoea in the vaccinated population compared to the unvaccinated population stratified by geographical location (regional, remote, very remote).
*N. meningitidis*—carriage	
6. Estimate the difference in carriage prevalence of disease-causing genogroups of *N. meningitidis* (A, B, C, E, X, W, Y) in 14–19-year-olds at baseline compared to 12 months after receiving two doses of 4CmenB.	Detection of all disease-causing genogroup of *N. meningitidis* (A, B, C, E, X, W, Y) by PCR at baseline and at 12 months.
7. Estimate the difference in carriage prevalence of each disease-causing genogroup of *N. meningitidis* (A, B, C, E, X, W, Y) in 14–19-year-olds at baseline compared to 12 months after receiving two doses of 4CmenB.	Detection of each disease-causing genogroup of *N. meningitidis* (A, B, C, E, X, W, Y) by PCR at baseline and at 12 months.
8. Estimate the difference in carriage prevalence of non-groupable *N. meningitidis* in 14–19-year-olds at baseline compared to 12 months after receiving two doses of 4CMenB.	Detection of all non-groupable *N. meningitidis* by PCR at baseline and at 12 months.
9. Describe the acquisition of all and invasive *N. meningitidis* after receiving two doses of 4CMenB.	Acquisition of *N. meningitidis* (negative at baseline, positive at 12-month follow-up) as measured by PCR.
Meningococcal disease	
10. Estimate the vaccine impact and effectiveness of 4CMenB administered to young adults on group B IMD in 14- to 19-year-olds	Notifications of group B IMD in the 4CMenB vaccinated population compared to the unvaccinated population.
	Notifications of IMD in the six years preceding 4CMenB vaccination compared to three years post vaccination (Interrupted time series analysis).
11. Estimate the vaccine impact and effectiveness of 4CMenB administered to young adults on group B IMD in all ages	4CMenB vaccination status in the population with IMD, compared to the general population (Screening method).
	4CMenB vaccination status in the population with IMD, compared to randomly selected controls from the Australian Immunisation Register (Case–control method).
Risk factors	
12. Identify characteristics associated with gonorrhoea in 15–19-year-olds.	Risk factors associated with gonorrhoea in 14–19-year-olds.
13. Identify characteristics associated with carriage of all genogroups of *N. meningitidis* in 14–19-year-olds at baseline and 12 months.	Risk factors associated with carriage of all genogroups of *N. meningitidis* in 14–19-year-olds at baseline and 12 months.
14. Identify characteristics associated with carriage of disease-causing genogroups of *N. meningitidis* (A, B, C, E, W, X, Y) in 14–19-year-olds at baseline and 12 months.	Risk factors associated with carriage of disease-causing genogroups of *N. meningitidis* (A, B, C, E, W, X, Y) in 14–19-year-olds at baseline and 12 months.
Cost effectiveness	
15. Estimate the cost effectiveness of a 4CMenB program for young adults on meningococcal disease and gonorrhoea.	Cost of meningococcal disease (acute care and management of sequelae up to one year) and cost of treatment of gonorrhoea compared to cost of 4CMenB vaccine.
Exploratory	
16. Describe the genetic diversity of *N. meningitidis* (disease associated and all carriage isolates).	Describe whole genome sequencing of *N. meningitidis* (A, B, C, E, W, X, Y) isolates.
17. Describe the genetic diversity of *N. gonorrhoea* isolates during implementation and follow up period.	Describe whole genome sequencing of *N. gonorrhoeae* isolates.

## Data Availability

The full study protocol is available from the corresponding author (H.S.M.) upon reasonable request and ethics approval.

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
