# Peer review of "An Observational Study to Assess the Effectiveness of 4CMenB against Meningococcal Disease and Carriage and Gonorrhea in Adolescents in the Northern Territory, Australia—Study Protocol"

_vaccines, 2022, doi:10.3390/vaccines10020309_

Round 1

Reviewer 1 Report

The manuscript describing the protocol to assess the effectiveness of 4CMenB 2 against meningococcal disease and carriage and gonorrhea in 3 adolescents in the Northern Territory, Australia is well written and described. In the study population section, the authors have mentioned that the informed consent will be obtained but please also include how the questions before and after vaccination, the participants may have will be addressed. If there will be any side effect(s) after vaccination whom the participant will contact/report. how it will be taken care of? How the information collected from various sources will be pooled? How the investigators will make it sure that they are getting the feedback? will there be any telephonic conversation in between?

Author Response

The manuscript describing the protocol to assess the effectiveness of 4CMenB 2 against meningococcal disease and carriage and gonorrhea in 3 adolescents in the Northern Territory, Australia is well written and described.

  1. In the study population section, the authors have mentioned that the informed consent will be obtained but please also include how the questions before and after vaccination, the participants may have will be addressed.

The participants and parents are provided with contact details of the study coordinators who will provide any additional information. We have revised the manuscript as “Participants and parents are provided with the contact details of the study coordinators to obtain any additional information that they may require prior to or after receiving the vaccination” (page 5, lines 127-129).

  1. If there will be any side effect(s) after vaccination whom the participant will contact/report. how it will be taken care of?

The current Northern Territory reporting system for adverse events following immunisation (AEFI) will be followed during the study.

We have added the following sentence to the section on study safety monitoring and surveillance “Participants/parents are given the contact details of the Immunisation Unit of the NT Centre for Disease Control and advised to report any AEFI to the Immunisation Unit” (page 7, lines 229-231).

  1. How the information collected from various sources will be pooled?

We have added the following sentence to the data and statistical analysis plan section “For the case-control and screening method analyses, data custodians from AIR and CDC NT will link participant records and assign unique study IDs prior to sending data to the study team. No identifying information will be stored in the main study database. Infection rates for IMD and gonorrhoea for various demographic subgroups will be estimated from CDC NT surveillance data, and a many-to-one merge will be used to link background rates to individual records, so as to adjust for these rates in the subsequent analyses. ” (page 8, lines 262-266).

  1. How the investigators will make it sure that they are getting the feedback? Will there be any telephonic conversation in between?

Participants and parents can provide feedback to the investigator team using the study facebook page or via contacting the study coordinators. We have added the following sentence to the section on patient and public involvement “Participants and parents are encouraged to provide feedback on the study using the B Part of it NT Facebook page” (page 7, lines 212-213).

We have also added a sentence to describe the adverse event reporting process as indicated above.

Reviewer 2 Report

Interesting protocol that deserves publication. Results can have relevant impact in publich health.

The authors describe a high incidence of gonorrhea in the region that may be prevented if the MenB vaccine provides crossed immunity. The evidence of crossed protection of the 4CMenB vaccine against N gonorrhea is recent and deserves field testing, as proposed in this protocol. The evidence of crossed protection of the 4CMenB vaccine against N gonorrhea is recent and deserves field testing, as proposed in this protocol.

Author Response

Interesting protocol that deserves publication. Results can have relevant impact in publich health.The authors describe a high incidence of gonorrhea in the region that may be prevented if the MenB vaccine provides crossed immunity. The evidence of crossed protection of the 4CMenB vaccine against N gonorrhea is recent and deserves field testing, as proposed in this protocol. The evidence of crossed protection of the 4CMenB vaccine against N gonorrhea is recent and deserves field testing, as proposed in this protocol.

We thank the reviewer for the positive comments.

Reviewer 3 Report

Thank you for your submission. 

The trial is addressing an important issue. It is a real-world observational study without a randomized control group, but the approach is sound given the public health context (i.e. it is essentially an opportunistic real-world trial).

The likely sample size at the conclusion of the study is clearly a key issue. What evidence was used to estimate the 40-50% uptake rate in this community?

Please check the references (e.g. 14,21). What is reference 28 and how do readers access it? This paper is highly relevant and should be mentioned (Ruiz Garcia Y, Sohn WY, Seib KL, et al. Looking beyond meningococcal B with the 4CMenB vaccine: the Neisseria effect. NPJ Vaccines 2021;6(1):130. doi: 10.1038/s41541-021-00388-3). It is also surprising that there is no mention of the clinical trials on the topic currently underway. 

Ruiz Garcia et al have mentioned these.

The first two paragraphs contain quite old data and references (e.g. a 2019 publication is referred to as being “recent”). Please update the information.

Author Response

The trial is addressing an important issue. It is a real-world observational study without a randomized control group, but the approach is sound given the public health context (i.e. it is essentially an opportunistic real-world trial).

  1. The likely sample size at the conclusion of the study is clearly a key issue. What evidence was used to estimate the 40-50% uptake rate in this community?

In a recent roll-out of a meningococcal ACWY vaccine program in the Northern Territory in response to an outbreak of serogroup W disease, around 50-60% of Aboriginal and Torres Strait Islander young people were vaccinated. We realise it is likely to be lower than this for a study but we are working with NT Health and have a wide range of communication strategies informed by our youth advisory group and Aboriginal Advisory group  to achieve this sample size. In a very similar study in South Australia assessing herd immunity of 4CMenB vaccine in adolescents we achieved 62% uptake in 15-18 year olds with 34,500 school students participating. Due to a more challenging environment in the Northern Territory we considered 40-50% a more realistic uptake.

  1. Please check the references (e.g. 14,21). What is reference 28 and how do readers access it?

We thank the reviewer and have now edited these references.

  1. This paper is highly relevant and should be mentioned (Ruiz Garcia Y, Sohn WY, Seib KL, et al. Looking beyond meningococcal B with the 4CMenB vaccine: the Neisseria effect. NPJ Vaccines 2021;6(1):130. doi: 10.1038/s41541-021-00388-3).

We thank the reviewer and have revised the paper to add this reference as “A recent review by Garcia et.al. provides a comprehensive summary of the potential effect of the 4CMenB vaccine against gonorrhoea (page 12, lines 435-437, reference 37).

  1. It is also surprising that there is no mention of the clinical trials on the topic currently underway. Ruiz Garcia et al have mentioned these.

We have now added a statement on clinical trial data as “At present, there are no data from randomized controlled trials on the effectiveness of meningococcal vaccines against gonorrhoea. However, data will be available in the near future from a few ongoing clinical studies from several countries (ClinicalTrials.gov:NCT04415424, NCT04350138, NCT04398849 and Marshall et al) [38].

  1. The first two paragraphs contain quite old data and references (e.g. a 2019 publication is referred to as being “recent”). Please update the information.

We agree with the reviewer and have now revised the manuscript to include more recent references.